# Influence of Polymers Surface Roughness on Noise Emissions in 3D-Printed UAV Propellers

**DOI:** 10.3390/polym17081015

**Published:** 2025-04-09

**Authors:** Florin Popișter, Horea Ștefan Goia, Paul Ciudin

**Affiliations:** Department of Design Engineering and Robotics, Faculty of Industrial Engineering, Robotics and Production Management, Technical University of Cluj-Napoca, B-dul Muncii 103-105, 400641 Cluj-Napoca, Romania; ciudin.oc.paul@student.utcluj.ro

**Keywords:** roughness, 3D printing, propellers, noise, aeroacoustics, UAV

## Abstract

Following the rising popularity of Unmanned Aerial Vehicles (UAVs) among large-scale users, in the form of domestic as well as professional drones, with applications in domains such as safety (e.g., surveillance drones), terrain mapping (using geo-scanning UAVs), videography drones, and high performance drones used in FPV (First Person View) drone competitions—as well as the rising wide accessibility of Fused Filament Fabrication (FFF)—especially considering the recent apparition and popularization of 3D printers capable of displaying exponential increases in performance metrics, the present work takes into consideration the practice of fabricating UAV propellers by means of FFF, focusing on the theoretical, as well as on the practical aspects of the roughness and quality observed at the level of the resulting surfaces. The paper proposes a set of propeller configurations obtained by combining popular propeller geometries, such as the Gemfan 51466-3 three-bladed propeller and the novel Toroidal propeller model, with a range of different fabrication materials, such as the Polyethylene Terephthalate Glycol (PETG) filament and the Polylactic Acid (PLA) filament. The main aim of the study is to reveal observations on the influence that the surface quality has on the performance metrics of a propeller. Based on the practical work, which aims to develop a comparative study between two drone propeller geometries manufactured by a nonconventional process, 3D printing, the practical applications in the study were carried out using low-cost equipment in order to evaluate the results obtained in a domestic setting. The study involves the identification of the noise values produced by the two geometries due to the roughness of the propeller surfaces.

## 1. Introduction

The increasing adoption of Unmanned Aerial Vehicles (UAVs) in urban environments has highlighted the necessity of mitigating noise emissions, particularly those generated by propellers. Surface roughness, a common consequence of the manufacturing process, is a significant contributor to aerodynamic noise. This study examines the impact of surface roughness in 3D-printed UAV propellers on noise emissions, with a particular emphasis on the aerodynamic and acoustic implications of manufacturing-induced surface irregularities.

Surface roughness plays a critical role in determining the aerodynamic performance of a UAV propeller. Irregularities on the blade surface disrupt the boundary layer flow, accelerating the transition from laminar to turbulent flow and thereby increasing both tonal and broadband noise emissions. Prior research has demonstrated that rough surfaces contribute to enhanced aerodynamic drag and intensified turbulent interactions [1,2]. Given that UAVs typically operate at low Reynolds numbers, they are particularly susceptible to these effects.

Additive manufacturing, particularly Fused Deposition Modeling (FDM) has become widely utilized for UAV propeller prototyping due to its cost-effectiveness and design flexibility. However, these techniques inherently produce rougher surfaces compared to traditional manufacturing methods, potentially leading to increased noise emissions. Studies have demonstrated that the layer-by-layer fabrication process results in surface imperfections that influence airflow characteristics and exacerbate noise generation [3,4].

Material selection also plays a crucial role in UAV propeller performance. Polylactic Acid (PLA) and Polyethylene Terephthalate Glycol (PETG) are two commonly used 3D-printing materials, each with distinct mechanical properties. PLA is more brittle and susceptible to fracture under mechanical stress, whereas PETG offers enhanced flexibility and durability. These material characteristics influence not only the structural integrity of the propeller but also its aerodynamic efficiency and noise emissions [5,6,7,8].

Optimized propeller designs seek to enhance airflow efficiency and minimize vortex formation, thereby reducing aerodynamic noise. However, the elevated roughness likely enhances aerodynamic drag and promotes irregular airflow patterns, which in turn exacerbate turbulent interactions and vortex formation at the blade surfaces [9,10,11,12].

The acoustic signature of 3D-printed propellers refers to the unique sound profile generated by the propeller during operation, which is influenced by several factors such as material choice, surface finish, design features, and manufacturing process. A notable distinction emerges when comparing the acoustic signatures of 3D-printed and traditionally manufactured propellers. The inherent characteristics of 3D printing, including surface roughness, material inconsistencies, and layer bonding patterns, contribute to this discrepancy [13,14,15].

Ensuring UAV safety is paramount, particularly in environments where drones operate in close proximity to humans or fragile infrastructure. Conventional propeller geometries, characterized by open-blade designs with high rotational speeds, pose a significant risk of injury or structural damage upon collision. These traditional configurations, while aerodynamically efficient, generate high impact forces due to their rigid structure and exposed blade edges. In contrast, toroidal propellers have emerged as a safer alternative, featuring a continuous loop design that reduces the likelihood of lacerations and mitigates the severity of collisions. The closed-loop architecture not only enhances safety by distributing impact forces more evenly but also offers additional aerodynamic benefits, such as improved thrust efficiency and reduced noise emissions. Compared to classical propellers, which can induce strong vortex shedding and turbulent interactions at the blade tips, toroidal designs demonstrate superior flow characteristics that contribute to quieter and more stable operation [16,17].

Studies presented in [18] are referring to the impact of the build edge profile in fused filament fabrication (FFF) on the surface roughness of 3D-printed parts. It focuses on predicting the surface finish based on the edge profile and provides insights into how various printing parameters and strategies influence the quality of the printed surface. The findings are particularly valuable for enhancing the accuracy and finish of 3D-printed components, especially in applications such as UAV propellers, where surface roughness plays a critical role in both aerodynamic and acoustic performance.

The literature survey incorporates more recent studies on UAV noise emissions, the effects of noise on humans, and the impact of surface roughness on noise generation. The revised state-of-the-art literature also highlights relevant quantitative results to provide a more comprehensive understanding of these topics [19,20,21,22,23,24,25].

The present paper refers to a conducted study which investigates how varying degrees of surface roughness in 3D-printed UAV propellers impact noise emissions. By analyzing the acoustic effects of manufacturing-induced surface variations and evaluating different materials such as PETG filament and PLA filament, this research aims to contribute to the development of UAV propellers with improved aerodynamic performance and reduced noise emissions. The findings have significant implications for UAV applications in commercial, urban, and defense sectors, where noise reduction is a critical design consideration.

## 2. Materials and Methods

Studying the influence of geometry and material and surface roughness parameters on a propeller’s performance in terms of generated sound imposes the necessity of isolating variables responsible for directly influencing the mentioned characteristics. The study thus proposes an echelon of distinct propellers, varying in terms of the FFF-specific polymers selected for fabrication, geometry, and processing parameters. Consequently, eight distinct propellers configurations were developed, as follows:

The nonconventional manufacturing process, 3D printing, was carried out using commercial printers and software that are low cost and widely available around the world. All the printing parameters were taken according to the manufacturing software capabilities, without any extra intervention. The comparison within the study concerned the results of the best quality and the standard quality of parts. The quality parameter, as shown in Table 1, varies between two configurations, including distinct parameter values, the so-called Standard and Best quality configurations. The two configurations present distinct values for the Layer Height (Standard: 0.2 mm, Best: 0.1 mm), and Line Width (Standard: 0.2 mm, Best: 0.1 mm) parameters, as presented in Table 2, among adjacent relevant post-processing parameters. The values were selected to be achievable using 3D printers demonstrating average performance capabilities.

The digital parametrization step undertaken prior to the manufacturing process of the propellers can be observed in Figure 1, which depicts a simulation of two of the specimens, revealing the selected geometric models.

Fused Filament Fabrication is a widely used 3D-printing method, using polymer-based filaments for constructing parts layer-by-layer. Among the polymers used in FFF, the Polylactic Acid (PLA) and Polyethylene Terephthalate Glycol (PETG) are frequently chosen, due to being widely available on the market, offering a low level of printing difficulty, and the good structural properties of their resulting parts. The ease-of-use characteristic of the chosen polymers is especially relevant for studying the noise emitted by propellers, as it facilitates the manufacturing of parts with refined and controlled surface finish parameters. 

Considering geometrical and structural stability, all the probe specimens (presented in Figure 2) have been acclimatized to the laboratory conditions for a minimum of 48 h after the end of the manufacturing process. This regulating step is taken to ensure the integrity of the specimens during the trials, and that the experimental processes do not affect the integrity of the probes in any manner, prioritizing the authenticity of the results.

The sound signature a propeller generates during exploitation is a metric of pronounced interest in the field. A propeller’s geometry is the main factor considered to affect the noise generated during use. The thesis stating that, at high speeds, the surface roughness of the propellers represents a notable source of noise as well, is tested by analyzing the differences in surface roughness measured across the proposed specimens.

Aside from correlating the surface roughness values measured for each propeller specimen, the trial was conducted to observe notable data identifying the relationships between the surface quality and the employed polymer, respectively, and the amplitude at which the process parametrization influences surface quality.

To ensure the accuracy and validity of the measurements, a high quality, dedicated roughness meter, the Insize ISR-C300 (INSIZE EUROPE S.L., Derio, Spain), was used in the process of measuring the roughness of the surface specimens. The measurement area for each geometry was chosen specifically as the one in which the airflow around the blade faced the most resistance, while ensuring that the measurements were carried out perpendicularly to the layer lines across the measured surface, as can be observed in Figure 3. The same image depicts the position in which the specimens are mounted during measurements, according to the propeller geometry.

The initial setup of the roughness meter includes, in accordance with ISO 16610-211 [26], the use of a Gaussian noise filter (INSIZE EUROPE S.L., Derio, Spain), applied on a cut-off length of 2.5 mm. The total evaluation length is set to 5 mm, the value allowed by the geometry, excluding measurement abnormalities caused by the non-planar geometry of the propellers. The area upon which the roughness meter’s probe acts in the measurement process presents an acceptably low grade of planar deviation, which, combined with the relatively short evaluation length, renders any abnormalities caused by geometrical inconsistency insignificant in the case of both proposed propeller geometries.

The measurement consists of a series of successive steps, conducted for each of the eight probes, as follows:Propeller specimen number one is mounted on the support.The spatial position and orientation of the specimen, relative to the roughness meter, is adjusted for the probe of the roughness meter to act upon the designated surface.A total of 10 measurements are carried out for the first blade of the propeller and the measured data are registered as the arithmetic mean of the values.The propeller is reoriented around its central axis, guiding the second blade to the measurement position.The measurement is carried out for the current blade and the measured data are collected.Steps 5 and 6 are repeated for the third and final blade of the propeller specimen.The steps are repeated for the remaining propeller specimens.

The process of measuring the sound signatures of each proposed propeller specimen imposes simulating authentic exploitation conditions of the propeller configurations, while isolating and accurately capturing factors of sound.

To ensure the precise quantification of noise emission, the experimental measurement stand, presented in Figure 4, must fulfill several critical functions. The propulsion system is simulated using a brushless DC motor specifically designed for UAV applications. The selected model, MT2204 (LIGPOWER-HQ, Nanchang, China), operates with a velocity constant (KV) of 2300 under an input voltage of 7.4 V, resulting in a calculated maximum theoretical rotational speed of 17,020 RPM. The motor operates at 60% of its maximum RPM to compare thrust across six propeller configurations. Rather than maximizing thrust, the focus is on relative performance differences among the propeller specimens.

To eliminate potential sources of unwanted noise, such as vibrations, the measurement stand is constructed with a robust design incorporating vibration-damping materials at all contact interfaces between its structural components. Furthermore, during measurements, the stand is enclosed within a 40 mm thick acoustic shielding composed of two layers of foam, effectively mitigating external noise interference. An additional two-layer foam structure is positioned beneath the entire setup to further minimize vibrational disturbances and enhance measurement accuracy. In order to ensure proper laboratory conditions for noise measurements, specific factors were considered and optimized. These factors included minimizing background noise in accordance with ISO 3382-1:2009 [27]. The process is monitored externally using a high-resolution camera and an additional light source. To accurately capture the sound signature of each propeller and analyze subtle differences in noise emissions, the trials employed a PCE-322A (PCE Instruments—PCE Deutschland GmbH, Meschede, Germany) professional sonometer positioned near the propellers, presenting a frequency range between 31.5 Hz and 8 kHz, and a resolution of 0.1 dB, with ±1.4 dB of accuracy.

The BLDC motor controller (Infineon Technologies AG, Augsburg, Germany), video camera and the PCE-322A sonometer are connected to a computer and the light source is turned on.The foam enclosure is placed over the experimental stand and the background noise is registered.The foam enclosure is lifted and propeller specimen number one is secured on the motor shaft.The foam enclosure is placed over the measurement stand.The process is monitored and recorded externally via the video stream.The sound data acquisition procedure is initiated.The BLDC motor is initiated, gradually reaching the imposed rotational speed value and stabilized.After stabilization, the sonometric measurement procedure, BLDC motor, and recording are stopped.The cover of the enclosure is lifted.The measured specimen is unmounted.Steps 3 through 10 are repeated for the remaining specimens.

## 3. Results

The precision of the results is contingent upon measurement precision, material consistency, and environmental conditions. Despite the implementation of standardized procedures and the utilization of calibrated instruments, residual uncertainties persist.

Surface roughness evaluation is subject to variability due to extrusion and cooling rates during FDM, even when identical printing parameters are employed. The ISR-C300 roughness meter’s 0.001 μm resolution introduces the possibility of deviations. The reliability of the results could be enhanced by testing multiple specimens under the same conditions, thereby reducing random errors. Furthermore, discrepancies in filament composition amongst manufacturers have been demonstrated to influence polymer flow and surface finish, thereby introducing additional variability in roughness measurements.

A similar situation pertains to sound measurements, which are similarly affected by uncertainties. Variations in background noise, even in controlled environments, can introduce variations. Propeller-specific vibrations at different frequencies complicate the isolation of aerodynamic noise. The PCE-322A sound level meter has a resolution limit of 0.1 dB. Furthermore, it should be noted that laboratory conditions are not necessarily representative of real-world factors such as wind turbulence and operational constraints. Conducting tests at different rotational speeds could enhance the robustness of noise analysis.

The way roughness is influenced by the different parameters of the probes, the polymeric material, and manufacturing parameters, is revealed by data resulting from roughness measured across the propeller specimen configurations, as revealed in Table 3, as values of Arithmetic Mean Roughness.

The analysis of surface roughness measurements reveals a consistent trend in which PLA exhibits higher arithmetic mean roughness values compared to PETG across all tested configurations. This trend is particularly evident in the Standard (S) quality class specimens, where C_PLA_S displays roughness values ranging from 16.206 to 17.638 μm, while C_PETG_S remains significantly lower, between 9.042 and 10.757 μm. A similar pattern is observed in the toroidal configurations, with T_PLA_S (15.248–15.849 μm) maintaining higher roughness than T_PETG_S (14.184–15.420 μm). The observed difference in surface roughness is likely attributed to the material properties of PLA and PETG, as PLA’s increased brittleness and lower elasticity may lead to greater surface irregularities during FFF. Conversely, PETG’s higher flexibility and reduced susceptibility to brittle fracture may contribute to a smoother deposited surface, resulting in lower overall roughness values. These variations in roughness have the potential to influence the noise characteristics of 3D-printed propellers.

The influence of propeller geometry on surface roughness is evident, with toroidal configurations consistently exhibiting higher roughness values than their classic counterparts for the same material and parametrization conditions. This trend is particularly noticeable in PETG-based propellers, where T_PETG_S presents roughness values between 14.184 and 15.420 μm, compared to the significantly lower range of 9.042 to 10.757 μm observed for C_PETG_S. A similar pattern is observed in the Best (B) quality class, where T_PETG_B (10.324–12.059 μm) remains rougher than C_PETG_B (5.847–8.663 μm). The increased roughness of toroidal propellers is probably due to their complex geometry. More pronounced curves in the vertical direction expose a greater number of layers, making it more difficult to achieve surface uniformity by means of layer-by-layer deposition.

The impact of surface quality on roughness measurements is evident, with Best quality instances consistently exhibiting lower arithmetic mean roughness values than Standard configurations across all tested propeller geometries and materials. This trend is particularly pronounced in toroidal PETG propellers, where T_PETG_B (10.324–12.059 μm) demonstrates a smoother surface compared to T_PETG_S (14.184–15.420 μm). A similar pattern is observed in classic PETG propellers, where C_PETG_B (5.847–8.663 μm) achieves the lowest roughness values in the dataset, in contrast to C_PETG_S (9.042–10.757 μm).

Correlating the obtained measurements of surface roughness, propeller geometries, and polymers used in the fabrication process, while revealing their influence on the sound signatures produced by each specimen of the studied echelon of propellers, necessitates analysis considering the results of noise measurements realized for the instance of each specimen, in terms of sound wave intensity (measured in decibels). The measurements are presented in Figure 5.

The relevant data, specifically the values of noise registered after the stabilization of the propeller specimens at the specified rotational speed values, are compiled in Table 4.

Focusing on materials, the C_PETG_B instance exhibits superior behavior in terms of noise, reaching lower decibel levels than the equivalent PLA specimen, the C_PLA_B. The majority of the PETG instances measure higher noise levels in comparison to their PLA counterparts, with the most pronounced discrepancy observable in the case of the C_PETG_S, C_PLA_S pair, and the C_PETG_S, which demonstrates the highest level of noise emission.

Propeller geometry plays a crucial role in determining noise characteristics. Toroidal propellers exhibit generally lower noise levels than classic geometry instances. The smoothening effect of the toroidal shape, which enhances airflow distribution and reduces tip vortex formation, appears to mitigate noise emissions. This is particularly evident for T_PLA_B and T_PETG_B, which demonstrates the lowest noise levels in the dataset. Conversely, classic PETG propellers (C_PETG_S and C_PLA_B) consistently produce the highest noise emissions, likely due to increased blade tip–air interactions and higher aerodynamic losses.

The established quality class directly correlates with noise levels. The class of Standard layer height and line width configurations generally results in higher noise emissions, as seen with C_PETG_S, in comparison to C_PETG_B. In contrast, Best surface quality configurations exhibit lower noise levels, particularly for C_PLA_B and T_PETG_B, reinforcing the hypothesis that smoother surfaces facilitate reduced aerodynamic drag and lower noise production. Additionally, T_PLA_B, which has one of the lowest roughness values, maintains one of the lowest noise levels, further emphasizing the importance of surface optimization in noise reduction strategies.

## 4. Conclusions

The study reveals that polymer material selection plays a central role in determining the surface quality of 3D-printed propellers and, consequently, their acoustic performance. Specifically, PLA-based propellers consistently exhibit higher arithmetic mean roughness (C_PLA_S: B1:17.638 [μm], B2:16.206 [μm], B3: 17.283 [μm]) compared to those fabricated from PETG (C_PETG_S: B1: 9.042 [μm], B2: 9.724 [μm], B3: 10.757 [μm]). This observation is likely attributable to the intrinsic material characteristics, where PLA’s brittleness leads to a tendency for surface irregularities, whereas PETG’s superior flexibility and resilience promote a smoother finish.

The investigation further demonstrates that the propeller geometry significantly influences the resulting surface roughness. The toroidal designs, characterized by their complex and curved profiles, consistently display higher roughness values (T_PETG_S: B1: 15.420 [μm], B2: 14.184 [μm], B3: 14.657 [μm]) when compared to classic geometries (C_PETG_S: B1:9.042 [μm], B2: 9.724 [μm], B3: 10.757 [μm]). This phenomenon is primarily due to the increased number of exposed layers in toroidal configurations, which complicates the layer-by-layer deposition process and results in greater surface irregularities. The model offers an aerodynamic advantage by promoting improved airflow and reducing vortex formation, appearing to outweigh the adverse effects of increased roughness. These benefits lead to lower noise emissions compared to classic propeller designs. Moreover, the study highlights that Best quality configurations, achieved through optimal pre-printing parameterization, are critical in reducing roughness, thereby reinforcing the importance of parameter optimization in the FFF process.

The data indicate that process parametrization exerts a direct influence on the acoustic signature of the propellers. Specifically, specimens produced under suboptimal printing parameters consistently demonstrate increased noise emissions (T_PETG_S: 73.6 [dB], T_PLA_S: 71.8 [dB], 86.6 [dB]). The elevated roughness likely enhances aerodynamic drag and promotes irregular airflow patterns, which in turn exacerbates turbulent interactions and vortex formation at the blade surfaces. Consequently, these aerodynamic disturbances manifest as higher decibel levels in the sound measurements, underscoring the critical role that meticulous control of surface finish plays in minimizing noise in UAV propellers. Surface roughness appears to be less effective than pre-print parametrization in mitigating noise, concurrently exerting an unexpected causality to the measured sound levels. PETG instances (T_PETG_S: 73.6 [dB], C_PETG_S: 86.6 [dB]), despite displaying lower roughness levels, appear to generate more pronounced peaks of noise pollution, in comparison to the PLA probes (T_PLA_S: 71.8 [dB], C_PLA_S: 70.6 [dB]). This unintuitive result could reveal the influence of incidental material-specific factors in a further, case-focused investigation.

Based on these findings, an effective strategy for minimizing noise emissions should prioritize optimization parameters in a hierarchical manner. First and foremost, the geometry of the propeller should be refined to maximize aerodynamic efficiency. Following this, rigorous control of surface quality through enhanced pre-printing parameterization is essential to further reduce roughness and its potential impact on noise production. Finally, material selection should be carefully considered, with a preference for PETG over PLA due to its inherent advantages in producing smoother surfaces. This comprehensive approach, which integrates geometric design, process optimization, and material properties, provides a robust framework for achieving superior acoustic performance in 3D-printed propeller applications.

## Figures and Tables

**Figure 1 polymers-17-01015-f001:**
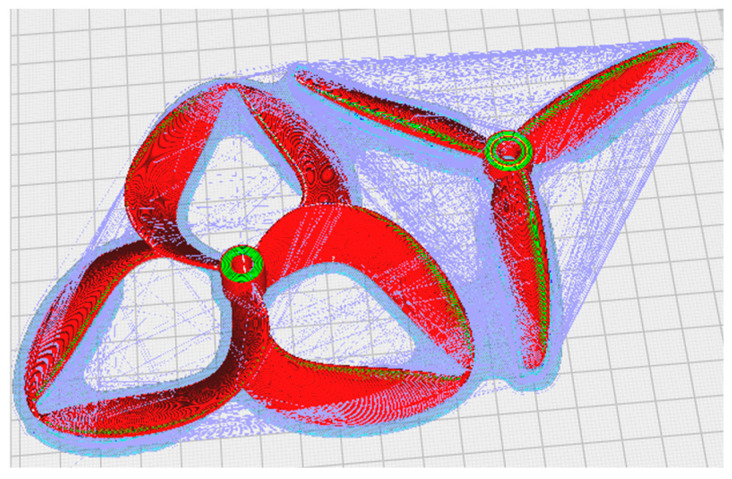
Pre-parametrization process of propeller probes.

**Figure 2 polymers-17-01015-f002:**
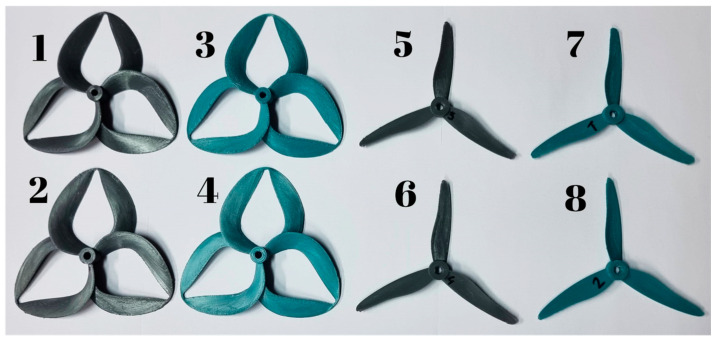
The 3D-printed propeller specimens.

**Figure 3 polymers-17-01015-f003:**
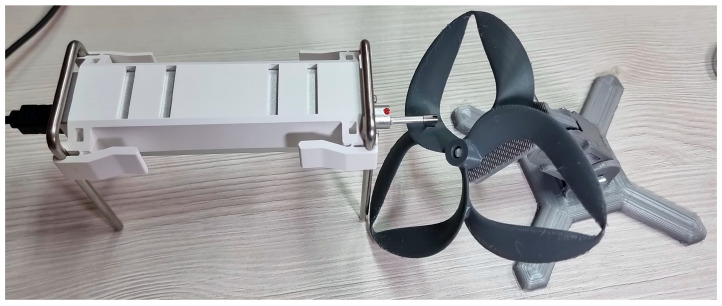
Roughness measurement process on one of the propeller specimens.

**Figure 4 polymers-17-01015-f004:**
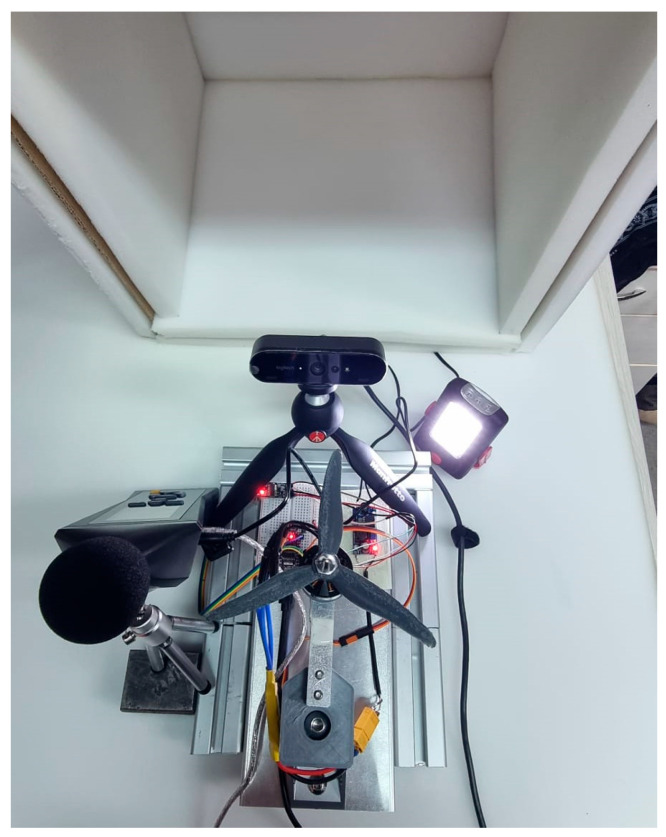
Sound measurement setup.

**Figure 5 polymers-17-01015-f005:**
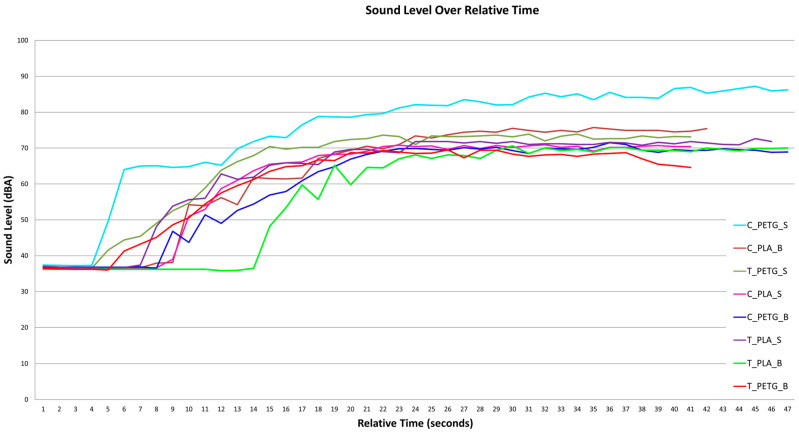
Graphical representation of the sound generated by the echelon of propeller specimens.

**Table 1 polymers-17-01015-t001:** Probes and constituting parameters.

Probe\Property	Geometry	Material	Quality
1. T_PETG_S	Toroidal	PETG	Standard
2. T_PETG_B	Toroidal	PETG	Best
3. T_PLA_S	Toroidal	PLA	Standard
4. T_PLA_B	Toroidal	PLA	Best
5. C_PETG_S	Classic 3 Bladed	PETG	Standard
6. C_PETG_B	Classic 3 Bladed	PETG	Best
7. C_PLA_S	Classic 3 Bladed	PLA	Standard
8. C_PLA_B	Classic 3 Bladed	PLA	Best

**Table 2 polymers-17-01015-t002:** Printing parameters and corresponding values of the probes.

Parameter	Material			
	PLA		PETG	
	Standard	Best	Standard	Best
Layer Height [mm]	0.1	0.2	0.1	0.2
Line Width [mm]	0.1	0.2	0.1	0.2
Printing Temperature [°C]	230 °C		245 °C	
Wall Line Count	6			
Infill Density [%]	100%			
Table Temperature [°C]	70 °C			
Nozzle diameter [mm]	0.4 mm			

**Table 3 polymers-17-01015-t003:** Arithmetic Mean Roughness measurements of the propeller specimens.

Specimen	Arithmetic Mean Roughness [μm]		
	Blade 1	Blade 2	Blade 3
T_PETG_S	15.420	14.184	14.657
T_PETG_B	11.394	10.324	12.059
T_PLA_S	15.849	15.248	15.683
T_PLA_B	9.852	10.783	9.908
C_PETG_S	9.042	9.724	10.757
C_PETG_B	8.663	5.847	7.247
C_PLA_S	17.638	16.206	17.283
C_PLA_B	9.463	9.201	8.762

**Table 4 polymers-17-01015-t004:** Measured levels of noise registered after stabilization.

Propeller Specimen	Noice Level [dB]
T_PETG_S	73.6
T_PETG_B	69.4
T_PLA_S	71.8
T_PLA_B	69.9
C_PETG_S	86.6
C_PETG_B	69.8
C_PLA_S	70.6
C_PLA_B	75.5

## Data Availability

The original contributions presented in this study are included in the article. Further inquiries can be directed to the corresponding authors.

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
