# Peer review of "Influence of Polymers Surface Roughness on Noise Emissions in 3D-Printed UAV Propellers"

_polymers, 2025, doi:10.3390/polym17081015_

Round 1
Reviewer 1 Report
Comments and Suggestions for Authors
This study addresses a timely and relevant topic in UAV development, focusing on the relationship between surface roughness in 3D-printed propellers and noise emissions. The work combines practical experimentation with material science and aerodynamics, offering insights into optimizing propeller design for reduced noise. While the methodology and results are promising, certain aspects require clarification and expansion to strengthen the scientific rigor and reproducibility of the findings.
1. The manuscript states that roughness measurements were taken "perpendicular to the layer lines" (Figure 3). However, How were measurements standardized across non-planar propeller surfaces?
2. Were multiple measurements taken per blade, and if so, how were outliers handled?
3. What was the frequency range of the sonometer (PCE-322A), and how was background noise quantified?
4. How was motor vibration noise isolated from aerodynamic noise?
5. The link between roughness and noise is discussed qualitatively. A deeper theoretical analysis (e.g., boundary layer transition, vortex shedding) would strengthen the conclusions.
6. How do the observed roughness values compare to critical thresholds for laminar-to-turbulent flow transition at UAV operating Reynolds numbers?
Author Response
Thank you for your time on reviewing our paper.

Reviewer 2 Report
Comments and Suggestions for Authors
Authors have proposed Influence of Polymers Surface Roughness on Noise Emissions in 3D-Printed UAV Propellers. The following are the observations/suggestions:
- The full form of FPV is not mentioned in the abstract. Ensure that the full form of all abbreviations used in the manuscript is included.
- The authors should highlight the novelty of their work in the abstract.
- Quantitative findings should be added to the abstract.
- The literature survey needs revision. Include more recent studies on UAV noise emission, the effects of noise on humans, the impact of surface roughness on noise, FFF on surface roughness, and the effect of 3D printing materials on surface roughness, along with discussions of quantitative results. The introduction depth should be enhanced further.
- A detailed discussion is required on the reasons for selecting parameters for this study, such as layer height (0.1 & 0.2 mm), line width (0.1 & 0.2 mm), etc. Include relevant references if necessary.
- The first paragraph of page 6 discusses thrust force measurement. The authors need to discuss the quantitative observations of thrust force for all eight specimens studied.
- Quality of Figure 5 should be improved; the continuous and dashed lines may be used to bifurcate between the specimens.
- Uncertainty analysis should be incorporated for both surface roughness and noise emission measurements.
- A quantitative comparison of noise emission results for all cases studied should be added.
- In the conclusions, the quantitative findings should be included. added.
Author Response

(The authors gave the same response as above.)

Round 2
Reviewer 2 Report
Comments and Suggestions for Authors
The comments are addressed.